# Knowledge, beliefs, attitudes and perceived risk about COVID-19 vaccine and determinants of COVID-19 vaccine acceptance in Bangladesh

Sultan Mahmud[1]*, Md. Mohsin[1], Ijaz Ahmed Khan[2], Ashraf Uddin Mian[3], Miah Akib Zaman[4]

**1** Institute of Statistical Research and Training, University of Dhaka, Dhaka, Bangladesh, **2** Innovations for Poverty Action Bangladesh (IPA-B), Dhaka, Bangladesh, **3** East West University, Dhaka, Bangladesh, **4** National Medical College, Dhaka, Bangladesh

* smahmud@isrt.ac.bd

**Data Availability Statement:** The data that support the findings of this study are openly available at a

## Abstract

Bangladesh govt. launched a nationwide vaccination drive against SARS-CoV-2 infection from early February 2021. The objectives of this study were to evaluate the acceptance of the COVID-19 vaccines and examine the factors associated with the acceptance in Bangladesh. In between January 30 to February 6, 2021, we conducted a web-based anonymous cross-sectional survey among the Bangladeshi general population. At the start of the survey, there was a detailed consent section that explained the study's intent, the types of questions we would ask, the anonymity of the study, and the study's voluntary nature. The survey only continued when a respondent consented, and the answers were provided by the respondents themselves. The multivariate logistic regression was used to identify the factors that influence the acceptance of the COVID-19 vaccination. A total of 605 eligible respondents took part in this survey (population size 1630046161 and required sample size 591) with an age range of 18 to 100. A large proportion of the respondents are aged less than 50 (82%) and male (62.15%). The majority of the respondents live in urban areas (60.83%). A total of 61.16% (370/605) of the respondents were willing to accept/take the COVID-19 vaccine. Among the accepted group, only 35.14% showed the willingness to take the COVID-19 vaccine immediately, while 64.86% would delay the vaccination until they are confirmed about the vaccine's efficacy and safety or COVID-19 becomes deadlier in Bangladesh. The regression results showed age, gender, location (urban/rural), level of education, income, perceived risk of being infected with COVID-19 in the future, perceived severity of infection, having previous vaccination experience after age 18, having higher knowledge about COVID-19 and vaccination were significantly associated with the acceptance of COVID-19 vaccines. The research reported a high prevalence of COVID-19 vaccine refusal and hesitancy in Bangladesh. To diminish the vaccine hesitancy and increase the uptake, the policymakers need to design a well-researched immunization strategy to remove the vaccination barriers. To improve vaccine acceptance among people, false

Open Science Framework https://doi.org/10.17605/
OSF.IO/B38HY.

**Funding:** The authors received no specific funding
for this work.

**Competing interests:** The authors have declared
that no competing interests exist.

rumors and misconceptions about the COVID-19 vaccines must be dispelled (especially on
the internet) and people must be exposed to the actual scientific facts.

## Introduction

The COVID-19 (the disease triggered by Severe Acute Respiratory Syndrome Coronavirus 2
(SARSCOV-2)) infection, declared as a "pandemic" by the World Health Organization, has
spread to all the countries of the world and as of February 20, 2021, infected more than 111
million people and claimed more than 2.4 million innocent lives [1]. The emergence of
COVID-19 has had a devastating effect on global healthcare systems with a ripple impact on
every aspect of human life as we experience it..With no proven treatments or medicines found,
governments across the world imposed border blackouts, travel bans, and quarantine [2] in a
bid to halt the spread of the virus that caused a voluminous economic downturn.. As of today,
the pandemic continues to ravage the world.

Scientists and researchers throughout the world have been working relentlessly to find a
way to get rid of the lethal disease. To combat communicable diseases, vaccines are considered
effective in developing a long-lasting immune system. About 2–3 million deaths per year are
avoided by vaccination [3]. In pandemics such as 1957, 1968, 1976, and 1977 outbreaks and
the H5N1 outbreak (1997–1998), and the 2009 H1N1 outbreak, many vaccines were developed
[4]. In the case of the COVID-19 pandemic, about 100 vaccines are in pre-clinical/clinical trials
and some of them have already been approved for mass inoculation [5]. With the approval of
vaccines for COVID-19, it is being expected that the pandemic can be controlled.

However, the development of vaccines is not the last thing to put an end to such a ubiqui-
tous and devastating virus. Scholars of vaccine hesitancy and adoption are warning policy-
makers and the scientific community that a successful vaccine is only the beginning. Based on
previous experiences with pandemic vaccines and vaccine hesitancy more broadly, a well-
researched strategy for rollout and adoption might be required in every country [6]. Most cur-
rent disease outbreak experience–the H1N1 outbreak in 2009, witnessed poor immunization
amongst adults, one study showing that 26 percent of refusers were worried about safety and
17 percent did not believe in the vaccine [7]. A recent study conducted in the USA revealed
that approximately 68% of all respondents are willing to get vaccinated against the COVID-19
with concerns about side effects and efficacy [8]. In a comprehensive survey of 19 nations con-
ducted in June 2020, 72% of participants suggested they were either likely or very likely to take
a vaccine, ranging from 89% in China to only 55% in Russia [9].

The question arises, "Why are people unwilling to get vaccinated against a devastating dis-
ease?". There is a plethora of research on the factors that influence vaccine uptake. Prior stud-
ies on seasonal and H1N1 influenza vaccinations have shown that vaccine attitudes and beliefs
are linked to vaccination intentions, which are a good predictor of vaccination uptake [10, 11].
According to one UK-based study, higher vaccination intentions were linked to the belief that
the COVID-19 disease would last a lot longer, while lower vaccination intentions were linked
to the belief that the dangers of COVID-19 had been inflated by the media [12]. Another study
examined the links between vaccine intention and sociodemographic characteristics, conclud-
ing that lower vaccination intention was connected to younger age and Black and minority
ethnicity [13]. However, the factors affecting vaccine intention and uptake might vary substan-
tially by territory, culture, and socioeconomic conditions.

In Bangladesh, from the very beginning of the pandemic, a substantial amount of unaware-
ness, rumors, and misinformation among general people about COVID-19 have been reported

[14]. It is also expected that there might be considerable misinformation and hesitancy in taking COVID-19 vaccines. Bangladesh govt. has revised its plan to inoculate 3.5 million instead of 6 million in February 2021 due to lukewarm response to online registration for COVID-19 vaccination [15]. It also relaxed the age limit to 40 years from previously stated 55 years for general people. This points to the unwillingness of the people of Bangladesh about receiving a COVID-19 vaccine. Bangladesh govt. has ordered and paid for at least 30 million doses of Oxford-AstraZeneca vaccine that will be delivered in installments across 2021 and also will receive another 68 million shots under the Covax initiative [16], led by the World Health Organization and Gavi, the Vaccine Alliance [15].

The Bangladesh govt has done a tremendous job in securing a good amount of shots of vaccines. Now, the challenge is to persuade people to take the vaccines. This study aims at investigating the knowledge, attitudes, and intention of people towards COVID-19 vaccines and the factors associated with COVID-19 vaccine acceptance in Bangladesh. For these purposes, we conducted an online anonymous survey between January 30 to February 6, 2021. The importance of conducting such a study in context to Bangladesh cannot be stressed enough as it would act as a guide for the Bangladesh govt. to encourage uptake among the general population.

## Methods

### Study design and study participants

In this study, we conducted a cross-sectional, web-based anonymous survey between January 30 to February 6, 2021. The participants were self-interviewed using an electronic questionnaire. At the start of the survey, there was a detailed consent section that explained the study's intent, the types of questions we would ask, the anonymity of the study, and the study's voluntary nature. The survey only continued when a respondent consented to the electronic informed consent, and the responses were provided by the respondents themselves. If consented, they were able to take part in the survey after an eligibility check step. One who was aged 18 or over and lived in Bangladesh was able to participate in this survey. The questionnaire was translated into Bangla, which was developed in English (see S1 Questionnaire). The respondents were informed that their participation was voluntary and after completing the survey they were requested to share the link with their contacts or acquaintances. Prior to circulating the questionnaire online, the questionnaire was validated and pilot tested. No sensitive/personal information was collected.

### Sample size

Since this study is aimed to examine the acceptability of the COVID-19 vaccine among the general population in Bangladesh and there was no previous literature from Bangladesh that examined the associated factors with this. We assumed that 50% of the general people have the factor of interest. And, we found a sample size of 591 using an online sample size calculator [17, 18] by assuming a 65% percent response rate, 5% precision or margin of error, and 50% proportion with a 95% confidence interval for the total population size of 1630046161 [19].

### Instruments

The researchers shared and advertised the KoBoToolbox online survey link to the public throughout the social network platforms of Facebook, WhatsApp, and Email. The questionnaire consisted of 5 sections: (i) Demographic background; (ii) Knowledge about COVID-19 and COVID-19 vaccination (iii) Belief and attitude (iv) Perceived barrier, perceived likelihood,

perceived severity (v) Vaccine acceptability. The questionnaire was based on previous research [20–22].

**Demographic background.** At the beginning of the survey, we completed the eligibility check by asking two questions "How old are you (in years)?" and "Do you currently live in Bangladesh?". This part of the questionnaire also contains personal details, including sex, religion, marital status, occupational status and monthly household income, and previous vaccination experience.

**Knowledge about COVID-19 and COVID-19 vaccination.** In this part, to measure the knowledge regarding COVID-19 and COVID-19 vaccination, participants were asked a series of yes/no questions. Such as "Is COVID-19 a lethal infectious disease?", "Is COVID-19 deadlier for elderly people (60+ years)?", "Do only elderly and sick people die of COVID-19?", "Can COVID-19 not spread from one to another by contact?", **"**Are hot and humid countries like Bangladesh safe from COVID-19?", "Is it human-made and deliberately released?", "Was the COVID-19 virus genetically engineered as part of a biological weapons program?", "Is this a normal disease like cold/cough and fever?", "Do people recover from it without any treatment?", "Is COVID-19 caused by the same virus that causes influenza (flu)?", "Testing can help people determine if they are infected with SARS-CoV-2, what do you think?", "Is there any effective medicine available for treating COVID-19/ coronavirus?".

**Belief and attitude.** In this part, we asked several questions with three options "Agree", "Neutral", "Disagree" to investigate the beliefs and attitudes of participants about COVID-19 and COVID-19 vaccination. For instance, the attitudes toward the COVID-19 and COVID-19 vaccination were inspected by asking "Vaccination is an effective way to prevent and control a disease", "Young (less than 30) and children do not need any vaccination against COVID 19", "We need to prioritize going back to our normal routines (opening schools, colleges, Office) as soon as possible by maintaining safety protocols", "It should be a crime if people know that they have COVID-19 but they don't isolate them", "The Covid-19 vaccines that are being inoculated worldwide are effective and safe", "Vaccines should be marketed and distributed entirely by the government in Bangladesh- what do you think?".

**Perceived berried, perceived likelihood, and perceived severity.** We cover perceived likelihood by asking "What do you think is the chance that you will get COVID-19 in the future?" with options "Low chance", "Medium chance" and "Higher chance". The perceived severity was also covered by asking "How severe do you think it would be if you get COVID-19?" with three options "Not at all/ low severe", "Medium severe", "Higher severe". The perceived barriers toward the COVID-19 and COVID-19 vaccination were inspected by asking two questions "If I decided to get the COVID-19 vaccine, it would be hard to find a provider or clinic that could give me the vaccine." With three options "Agree", "Neutral" and "Disagree" and "The COVID-19 vaccine might have side effects" with similar three options.

**Vaccine acceptability.** Vaccine acceptability was the main outcome of this study. Firstly the participants who chose "Yes" to the question "Have you heard of any vaccine that is going to be inoculated in Bangladesh?" were asked, "Bangladesh Govt. is going to inoculate COVID-19 vaccine, will you take it?". If the respondent chose "No" then we observed the reasons for not accepting the covid-19 vaccine by asking a multiple-choice question. Otherwise, we asked "When will you or your family members take the vaccine?" with the options "Will take as soon as possible" "After 2–6 months if seems safe and effective", "If COVID-19 becomes deadlier in Bangladesh", and "Not sure". We also observed the willingness to pay for COVID-19 vaccination by asking "What should be the price of a complete dose of a vaccine?" with three options "should be free", "1–1000", "1000+". We also observed perception about vaccination priority group by asking "Considering the current scenarios, who do you think should receive the first shipment of the vaccine in Bangladesh?" with options "Healthcare workers/professionals", "Elderly people (60+ years)", "People who have underlying diseases", "Politicians", "Others".

### Consent and ethical consideration

A voluntary online consent was taken by sharing a consent form on the timeline/inbox/WhatsApp of the respondents, which contained the outline of the research purpose and brief instructions regarding the survey.

### Statistical analysis

The acceptance of the COVID-19 vaccine is the primary outcome of this study. We classified respondents into two groups (accept group and refuse group) based on the response to the question "Bangladesh Govt. is going to inoculate COVID-19 vaccine, will you take it?". Among the accept group who chose "will take as soon as possible" to the question "when will you or your family member take it?" further classified into vaccine "immediate" group. And others who selected any of the options among "After 2–6 months if seems safe and effective", "If COVID-19 becomes deadlier in Bangladesh" and "Not sure" assigned to vaccine delay group [23]. We calculated the knowledge score for each of the participants based on the number of valid/correct answers for the 12 questions that were asked in the subsection Knowledge about COVID-19 and COVID-19 vaccination. Knowledge scores can vary from 0 to 12.

We have done the exploratory analysis/descriptive statistics (bivariate analysis, frequencies analysis, means, graphs, etc.) to inspect the socio-demographic characteristics, perceived barriers, perceived likelihood, perceived severity, willingness to pay, beliefs, and attitudes toward COVID-19 and COVID-19 vaccination. The chi-square test was performed to compare the baseline information among two groups (accept group and refuse group; immediate and delay group). The multivariate logistic regression was also performed to identify the influencing factors in decision-making for accepting the COVID-19 vaccine among both pairs of groups (accept group and refuse group; immediate and delay group). Logistic regression produced the odds ratio (OR), 95% confidence interval (95% CI), and P-value. The factors were considered to be endorsed in the regression models which showed a statistically significant correlation (at 10% level of significance) within the bivariate analysis.

## Results

Six hundred and forty-seven people opened the survey link. Among them, twenty-five people refused the survey, seventeen people were not eligible to complete the survey and six hundred and five people submitted the completed survey. The characteristics of participants are presented in Table 1. As desired, the sample is broadly representative of Bangladesh's general population (Islam 90.25%, Hinduism 8.26, Buddhism 0.83, and Christianity 0.66 [24]). The majority of the respondent tended to be aged between less than 50 (82.78%), and male (62.15%). More than half of the respondents live in urban areas (60.83%). A large proportion of respondents have a University degree (45.45%) while 26.94% are hon's running students and 27.60% of respondents passed HSC/Alim/ Vocational degree/Nursing or less. A small portion of the participants (18.68%) had previous vaccination experience after age 18.

Overall, 61.16% (370/605) of the respondents were classified as an accept group and among them, 35.14% (130/307) were willing to take it immediately, and the rest 47.84% (240/605) wanted to delay in taking a COVID-19 vaccine (see Fig 1). Most of the participants (71.74%) expressed that the COVID-19 vaccine should be free and the rest of the participants indicate they would like to pay out of pocket for a vaccine was (25.29%) Tk 1–1000, and (2.98%) more than 1000. To inspect the reasons behind the unwillingness of accepting the COVID-19 vaccine, we asked a question with multiple selection options. Among 235 participants who showed unwillingness to accept the COVID-19 vaccine, 78.52% were worried about the side effects or safety of the COVID-19 vaccine, 76.17% were doubtful about the efficacy of the

**Table 1. Participant characteristics (n = 605).**

| Variable | | Number | Percent |
|---|---|---|---|
| Age | | | |
| | In between 18 to 29 | 157 | 25.59 |
| | In between 30 to 50 | 346 | 57.19 |
| | In between 51 to 70 | 77 | 12.73 |
| | In between 71 to 100 | 25 | 4.13 |
| Gender | | | |
| | Male | 376 | 62.15 |
| | Female | 229 | 37.85 |
| Region where the respondent lives | | | |
| | Urban | 314 | 51.90 |
| | Rural | 291 | 48.10 |
| Marital status | | | |
| | Unmarried | 403 | 66.61 |
| | Married | 191 | 31.57 |
| | Divorced/Separated/Widowed | 11 | 1.82 |
| Highest qualification | | | |
| | HSC/Alim/ Vocational/Nursing or less | 167 | 27.60 |
| | University degree (Hon's/MBBS/ Masters or above) | 275 | 45.45 |
| | Hon's running | 163 | 26.94 |
| Religion | | | |
| | Islam | 546 | 90.25 |
| | Hindu | 50 | 8.26 |
| | Christian | 5 | 0.83 |
| | Buddhism | 4 | 0.66 |
| Monthly average household income | | | |
| | Less than 30,000 | 332 | 54.88 |
| | 30,000–39,999 | 82 | 13.55 |
| | 40,000–49,999 | 51 | 8.43 |
| | 50,000–74,999 | 68 | 11.24 |
| | 75,000 or over | 50 | 8.26 |
| | Don't know/unwilling to reveal | 22 | 3.64 |
| Employment status | | | |
| | Service holder (Govt./private) | 179 | 29.59 |
| | Entrepreneur/business | 148 | 24.46 |
| | Student | 218 | 29.92 |
| | Housewife/Retired/Unemployed/ Other[#] | 97 | 16.03 |
| Did you take any vaccine after 18 years of age? | | | |
| | No | 492 | 81.32 |
| | Yes | 113 | 18.68 |

# includes agriculture, retirement, intern, part-time job, autonomous organization.

COVID-19 vaccine. Some of the respondents (42%) were also doubtful about the COVID-19 Vaccine since it is coming from India (Oxford-AstraZeneca vaccine produced by Serum Institute, India). Almost 36% of respondents thought vaccination is not necessary since COVID-19 is going away or they are young (See Fig 2).

Participants were asked a series of yes/no questions to assess more general knowledge and belief about COVID-19 and COVID-19 Vaccination. The percentage of yes, no and don't

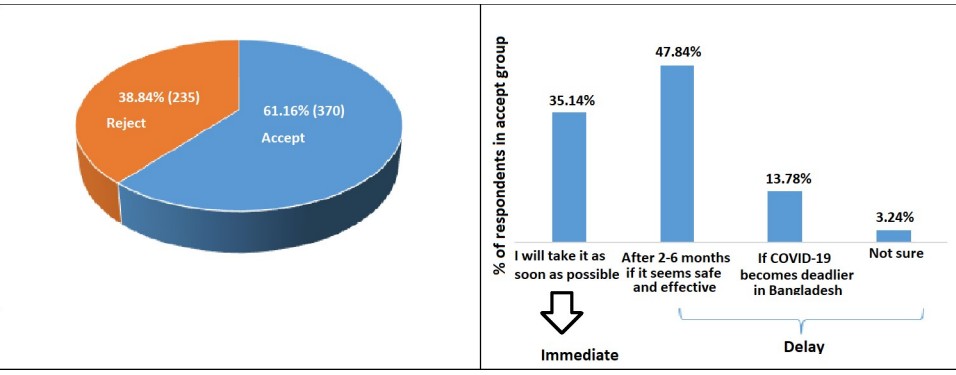

**Fig 1. The acceptance of COVID-19 vaccine in Bangladesh.**

know with the correct answer can be seen in Table 2. A large number of respondents (62.15%) think that COVID-19 is a lethal infectious disease. Almost half (51.40%) of respondents provided a positive answer with the question "*Is COVID-19 deadlier for elderly people (60+ years)*" and 84.63% of respondents provided the correct answer to the question "*Do only elderly and sick people die of COVID-19?*". Very few (30.51%) of the participants thought that COVID-19 is a normal disease like cold/cough and fever. The survey results showed overall 39.67% of respondents have good knowledge, 44.97% of respondents have medium knowledge and 16.36% have limited knowledge about COVID-19 and COVID-19 vaccination (Fig 3). Attitudes and beliefs towards vaccination, perceived barriers, and perceived risks were also inspected in Table 2. A large proportion of respondents (74.71%) agreed with the statement "*vaccination is an effective way to prevent and control a disease*" while 15.54% disagreed and 9.75% were neutral. A small proportion of 10.91% agreed with the statement "*Young (less than 30) and children do not need any vaccination against COVID 19*" while 54.55% disagreed and 34.55% were neutral. A large proportion (39.01%) of respondents were neutral with the statement "*The COVID-19 vaccines that are being inoculated worldwide are not effective and safe*" while 37.85% agreed and 23.14% disagreed. A lower portion (28.10%) of the respondents reported that it would not be hard to find a provider or clinic that could give him/her the

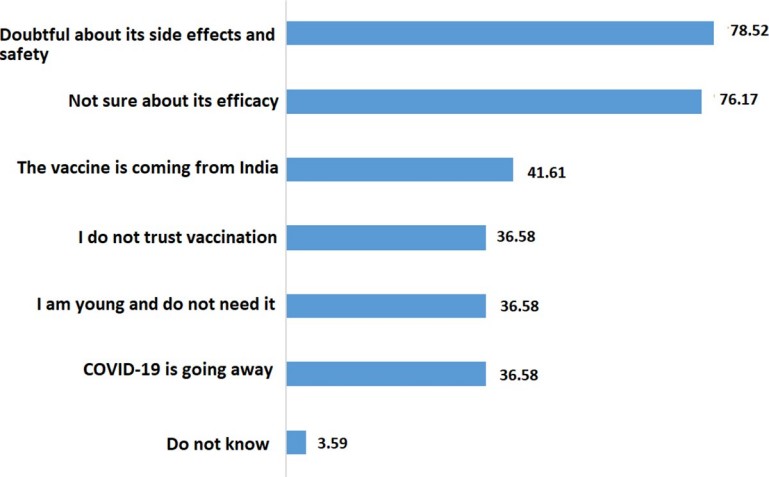

**Fig 2. Reasons behind the rejection of COVID-19 vaccine (n = 298).**

**Table 2. Descriptive statistics for items measuring knowledge, beliefs, and attitudes about COVID-19, COVID-19 vaccination (n = 605).**

| Knowledge and beliefs | Yes | No | Don't know |
|---|---|---|---|
| Is COVID-19 a lethal infectious disease? | **62.15 (376)** | 34.21 (207) | 3.64 (22) |
| Is COVID-19 deadlier for elderly people (60+ years)? | **51.40 (311)** | 46.45 (281) | 2.15 (13) |
| Do mostly elderly and sick people die of COVID-19? | 12.89 (78) | **84.63 (512)** | 2.48 (15) |
| Can COVID-19 not spread from one to another by contact? | 66.28 (401) | **30.91 (187)** | 2.81 (17) |
| Are hot and humid countries like Bangladesh safe from COVID-19? | 6.12 (37) | **90.74 (549)** | 3.14 (19) |
| Is it human-made and deliberately released? | 40 (40) | **89.92 (544)** | 3.47 (21) |
| Was the COVID-19 virus genetically engineered as part of a biological weapons program? | 6.78 (41) | **90.74 (549)** | 2.48 (15) |
| Is this a normal disease like cold/cough and fever? | 30.58 (185) | **66.94 (405)** | 2.48 (15) |
| Do people recover from it without any treatment? | 27.44 (166) | **68.93 (417)** | 3.64 (22) |
| Is COVID-19 caused by the same virus that causes influenza (flu)? | 27.60 (167) | **70.41 (426)** | 1.98 (12) |
| Testing can help people determine if they are infected with SARS-CoV-2, what do you think? | **67.93 (411)** | 28.93 (175) | 3.14 (19) |
| Is there any effective medicine available for treating COVID-19/ coronavirus? | 27.93 (169) | **43.31 (262)** | 28.76 (174) |
| **Attitude and beliefs** | **Agree** | **Neutral** | **Disagree** |
| Vaccination is an effective way to prevent and control a disease | 74.71 (452) | 9.75 (59) | 15.54 (94) |
| Young (less than 30) and children do not need any vaccination against COVID 19 | 10.91 (66) | 34.55 (209) | 54.55 (330) |
| We need to prioritize going back to our normal routines (opening schools, colleges, offices) as soon as possible by maintaining safety protocols. | 71.07 (430) | 15.37 (93) | 13.56 (82) |
| It should be a crime if people know that they have COVID-19 but they don't isolate them | 74.22 (449) | 11.07 (67) | 14.71 (89) |
| The Covid-19 vaccines that are being inoculated worldwide are effective and safe | 37.85 (229) | 39.01 (236) | 23.14 (140) |
| Vaccines should be marketed and distributed entirely by the government in Bangladesh- what do you think? | 84.3 (510) | 11.07 (67) | 4.63 (28) |
| **Perceived barriers** | **Agree** | **Not sure** | **Disagree** |
| If I decided to get the COVID-19 vaccine, it would be hard to find a provider or clinic that could give me the vaccine. | 42.31 (256) | 29.59 (179) | 28.10 (170) |
| The COVID-19 vaccine might have side effects. | 21.32 (129) | 38.35 (232) | 40.33 (244) |
| **Perceived risk** | **No or low** | **Medium** | **High** |
| What do you think is the chance that you will get COVID-19 in the future? | 34.05 (206) | 40.17 (243) | 25.79 (156) |
| How severe do you think it would be if you get COVID-19? | 40.17 (243) | 36.36 (220) | 23.47 (142) |

vaccine while 42.31% opposed the view. A small proportion of the respondents (34.05%) mentioned that there is a low chance to get COVID-19 in the future, while 40.17% mentioned medium chance, and 25.79% mentioned medium chance. Concerning perceived severity,

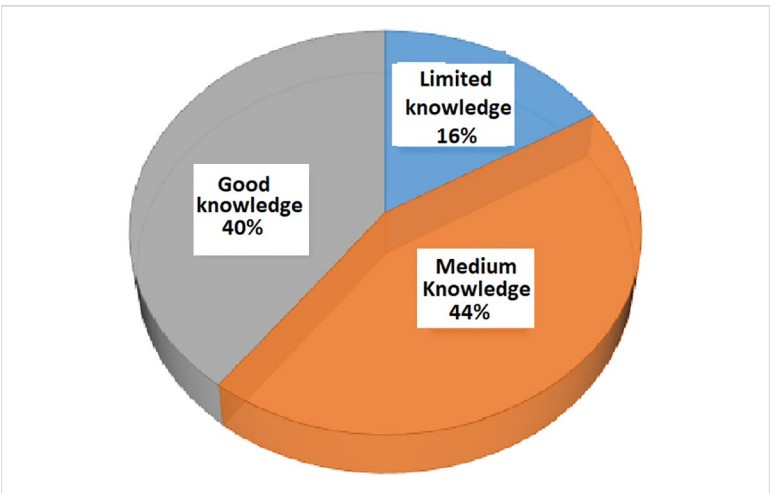

**Fig 3. Knowledge distribution of participants (knowledge about COVID-19 and COVID-19 vaccination.**

23.47% of respondents indicated that they would experience highly severe COVID-19 symptoms, 36.36%, and 40.17% indicated medium and low, respectively.

The results from multivariate logistic regressions are shown in Table 3. Parameter estimates of the multivariate logistic regression indicate the odds of occurring the event into the concerned categories of the predictor variable compared to the odds of occurring the event into the reference category of the same predictor variable. Here, the odds of accepting the vaccine is defined as the ratio of the probability of accepting the vaccine and the probability of rejecting the vaccine [25]. The model was based on 605 cases with complete data and showed an association between willingness to be vaccinated (yes/no) and predictor variables. Another model was based on 370 cases (vaccine accepted group) that explored the association between willingness to immediate vaccination and predictor variables. The potential predictors were trained in the model if they showed significant association (at a 10% level of significance) with the corresponding response variable in Table 4. The findings indicate that the odds of acceptance of the COVID-19 vaccine among female respondents is 62% lower than the male respondents. The respondents aged 30 to 50 and 51 to 70 had a 6.79 and 7.89 times higher (respectively) chance to accept the COVID-19 vaccine compared to respondents aged 18 to 29. The odds of accepting the COVID-19 vaccine among respondents who live in rural areas is 81% lower in comparison with the respondents who live in urban areas. The participants who have a university degree and who are hon's (undergrad) running students had 21.38 times higher odds of taking the COVID-19 vaccine compared to the participants who had a lower educational qualification (HSC/Alim/Vocational education/Nursing or less). Respondents with income 30.000 to 39,999 showed 4.35 times higher odds, income 40,000–49,999 had 8.95 times higher odds, income 50,000–74,999 displayed 8.44 times higher odds, and income 75,000 or over had 6.31 times higher odds to accept the COVID19 vaccine compared to respondents with income less than 30,000. Respondents with previous vaccination experience after the age of 18 years had 4.79 times higher odds compared to respondents who do not have any vaccination after age 18 years. People who disagreed with the statement *"the vaccines that are being inoculated worldwide are effective and safe"* had a 90% lower likelihood to accept the COVID-19 vaccine relative to those who agreed with it. Participants who believe they have a medium and a high chance to be infected with COVID-19 in the future showed respectively 3.19 times and 8.68 times higher

**Table 3. Results of the multivariate logistic regressions analyzing the associations with acceptance of vaccination and acceptance of immediate vaccination.**

| Variable | Levels | Willingness to be vaccinated | | Willingness to immediate vaccination | |
|---|---|---|---|---|---|
| | | OR (95% CI) | P-value | OR (95% CI) | P-value |
| Age | In between 18 to 29 | Ref | | Ref | |
| | In between 30 to 50 | 6.79 (2.63–17.55) | <0.001 | 0.64 (0.34–1.2) | NS |
| | In between 51 to 70 | 7.89 (2.05–30.39) | <0.003 | 0.8 (0.32–1.98) | NS |
| | In between 71 to 100 | 0.87 (0.13–5.98) | NS | 0.95 (0.16–5.62) | NS |
| Gender | Male | Ref | | Ref | |
| | Female | 0.38 (0.18–0.79) | 0.01 | 0.59 (0.34–1.03) | NS |
| Region where respondent lives | Urban | Ref | | Ref | |
| | Rural | 0.19 (0.09–0.4) | <0.001 | 2.03 (1.22–3.37) | 0.01 |
| Highest qualification | HSC/Alim/ Vocational/Nursing or less | Ref | | Ref | |
| | University degree (Hon's/MBBS/ Masters or above) | 21.38 (7.02–65.14) | <0.001 | 0.35 (0.14–0.85) | 0.02 |
| | Hon's running | 2.01 (0.72–5.59) | NS | 0.31 (0.12–0.84) | 0.02 |
| Religion | Islam | Not retained | | Ref | |
| | Hinduism | | | 3.79 (1.68–8.56) | 0.001 |
| | Buddhism | | | 1.42 (0.14–14.67) | NS |
| | Christian | | | | |
| Monthly average household income | Less than 30,000 | Ref | | Not retained | |
| | 30,000–39,999 | 4.35 (1.52–12.47) | 0.006 | | |
| | 40,000–49,999 | 8.95 (1.91–41.82) | 0.005 | | |
| | 50,000–74,999 | 8.44 (2.19–32.59) | <0.01 | | |
| | 75,000 or over | 6.3 (0.75–52.95) | NS | | |
| | Don't know | 1.25 (0.15–10.53) | NS | | |
| Marital Status | Unmarried | Not retained | | Ref | |
| | Married | | | 2.15 (1.22–3.8) | 0.01 |
| | Divorced/Separated/Widowed | | | 0.8 (0.09–7.05) | NS |
| Employment status | Service holder (govt/private) | Ref | | Not retained | |
| | Entrepreneur/business | 11.83 (3.48–40.25) | <0.001 | | |
| | Student | 2.3 (0.74–7.17) | NS | | |
| | Housewife/Retired/Unemployed/ Other# | 2.2 (0.66–7.29) | NS | | |

(*Continued*)

**Table 3.** (Continued)

| Variable | Levels | Willingness to be vaccinated | | Willingness to immediate vaccination | |
|---|---|---|---|---|---|
| | | OR (95% CI) | P-value | OR (95% CI) | P-value |
| Did you take any vaccine after 18 years of age? | No | Ref | | Not retained | |
| | Yes | 4.79 (1.65–13.9) | <0.001 | | |
| What do you think is the chance that you will get COVID-19 in the future? | No or low chance | Ref | | Not retained | |
| | Medium chance | 3.19 (1.29–7.9) | 0.01 | | |
| | High chance | 8.68 (2.87–26.27) | <0.001 | | |
| How severe do you think it would be if you get COVID-19? | No or low severe | Ref | | Not retained | |
| | medium severe | 1.02 (0.45–2.32) | NS | | |
| | Very severe | 4.47 (1.66–12.08) | <0.001 | | |
| We need to prioritize going back to our normal routines (opening schools, colleges, offices) as soon as possible by maintaining safety protocols. | Agree | Ref | | Not retained | |
| | Not sure | 2.46 (0.91–6.69) | NS | | |
| | Disagree | 2.55 (0.85–7.64) | NS | | |
| Have you heard of any vaccine (s) that have been approved globally for mass inoculation? | No | Ref | | Not retained | |
| | Yes | 1.57 (0.73–3.4) | NS | | |
| The vaccines that are being inoculated worldwide are effective and safe | Agree | Ref | | Ref | |
| | Neutral | 0.27 (0.11–0.67) | 0.01 | 0.44 (0.26–0.75) | <0.001 |
| | Disagree | 0.1 (0.04–0.29) | <0.001 | 0.32 (0.13–0.8) | 0.01 |
| If I decided to get the COVID-19 vaccine, it would be hard to find a provider or clinic that could give me the vaccine. | Agree | Ref | | Ref | |
| | Not sure | 1.79 (0.73–4.41) | NS | 0.61 (0.35–1.08) | NS |
| | Disagree | 0.05 (0.02–0.12) | <0.001 | 1.12 (0.56–2.25) | NS |
| The COVID-19 vaccine might have side effects. | Agree | Ref | | Not retained | |
| | Not sure | 3.64 (1.32–10.05) | 0.01 | | |
| | Disagree | 8.58 (3.15–23.35) | <0.001 | | |
| Knowledge score | Limited knowledge | Ref | | Not retained | |
| | Medium knowledge | 8.39 (1.96–35.97) | 0.004 | | |
| | Good knowledge | 22.23 (4.53–109.11) | <0.001 | | |

COVID-19 = coronavirus disease 2019; OR indicates Odds Ratio; CI = confidence interval; Ref = reference group; NS indicates a non significant result at 5% level; Not retained indicates that the corresponding predictor excluded from the model.

odds of taking the COVID-19 vaccine compare to the respondents who believe they have low or no chance to be infected with COVID-19 in future. Respondents who believe if they get infected with COVID-19 it would be highly severe demonstrated 4.47 times higher odds of

**Table 4. Association of different factors with the willingness to be vaccinated and willingness to immediate vaccination.**

| Variable | Levels | Willingness to be vaccinated | | | Willingness to immediate vaccination | | |
|---|---|---|---|---|---|---|---|
| | | No % (n) | Yes % (n) | P-value | delay % (n) | immediate % (n) | P-value |
| Age | In between 18 to 29 | 53.50 (84) | 46.50 (73) | <0.001 | 65.75 (48) | 34.25 (25) | 0.06 |
| | In between 30 to 50 | 30.06 (104) | 69.94 (242) | | 68.18 (165) | 31.82 (77) | |
| | In between 51 to 70 | 41.56 (32) | 58.44 (45) | | 48.89 (22) | 51.11 (23) | |
| | In between 71 to 100 | 60.00 (15) | 40.00 (10) | | 50.00 (5) | 50.00 (5) | |
| Gender | Male | 32.71 (123) | 67.29 (253) | <0.001 | 61.66 (156) | 38.34 (97) | 0.06 |
| | Female | 48.91 (112) | 51.09 (117) | | 71.79 (84) | 28.21 (33) | |
| Region where respondent lives | Urban | 21.66 (68) | 78.34 (246) | <0.001 | 71.14 (175) | 28.86 (71) | 0.003 |
| | Rural | 57.39 (167) | 42.61 (124) | | 52.42 (65) | 47.58 (59) | |
| Highest qualification | HSC/Alim/ Vocational/ Nursing or less | 82.63 (138) | 17.37 (29) | <0.001 | 37.93 (11) | 62.07 (18) | <0.001 |
| | University degree (Hon's/ MBBS/ Masters or above) | 11.64 (32) | 88.36 (243) | | 63.79 (155) | 36.21 (88) | |
| | Hon's running | 39.88 (65) | 60.12 (98) | | 75.51 (74) | 24.49 (24) | |
| Religion | Islam | 39.56 (216) | 60.44 (330) | NS | 67.27 (222) | 32.73 (108) | 0.01 |
| | Hindu | 34.00 (17) | 66.00 (33) | | 39.39 (13) | 60.61 (20) | |
| | Cristian | 20.00 (1) | 80.00 (4) | | 50 (2) | 50 (2) | |
| | Buddhism | 25.00 (1) | 75.00 (3) | | 100 (3) | 0 (0) | |
| Monthly average household income | Less than 30,000 | 47.29 (157) | 52.71 (175) | <0.001 | 65.14 (114) | 34.86 (61) | NS |
| | 30,000–39,999 | 26.83 (22) | 73.17 (60) | | 60.00 (36) | 40.00 (24) | |
| | 40,000–49,999 | 19.61 (10) | 80.39 (41) | | 70.73 (29) | 29.27 (12) | |
| | 50,000–74,999 | 29.41 (20) | 70.59 (48) | | 77.08 (37) | 22.92 (11) | |
| | 75,000 or over | 24 (12) | 76 (38) | | 55.26 (21) | 44.74 (17) | |
| | Don't know | 63.64 (14) | 36.36 (8) | | 37.50 (3) | 62.50 (5) | |
| Marital Status | Unmarried | 39.21 (158) | 60.79 (245) | NS | 71.43 (175) | 28.57 (70) | <0.001 |
| | Married | 38.22 (73) | 61.78 (118) | | 51.69 (61) | 48.31 (57) | |
| | Divorced/Separated/ Widowed | 36.36 (4) | 63.64 (7) | | 57.14 (4) | 42.86 (3) | |

(*Continued*)

**Table 4.** (Continued)

| Variable | Levels | Willingness to be vaccinated | | | Willingness to immediate vaccination | | |
|---|---|---|---|---|---|---|---|
| | | No % (n) | Yes % (n) | P-value | delay % (n) | immediate % (n) | P-value |
| Employment status | Service holder (govt/private) | 32.96 (59) | 67.04 (120) | <0.001 | 61.67 (74) | 38.33 (46) | NS |
| | Entrepreneur/business | 27.03 (40) | 72.97 (108) | | 64.81 (70) | 35.19 (38) | |
| | Student | 51.93 (94) | 48.07 (87) | | 73.56 (64) | 26.44 (23) | |
| | Housewife/Retired/ Unemployed/ Other# | 43.30 (42) | 56.70 (55) | | 58.18 (32) | 41.82 (23) | |
| Did you take any vaccine after 18 years of age? | No | 44.11 (217) | 55.89 (275) | <0.001 | 66.55 (183) | 33.45 (93) | NS |
| | Yes | 15.93 (18) | 84.07 (95) | | 60.00 (57) | 40.00 (38) | |
| What do you think is the chance that you will get COVID-19 in the future? | No or low chance | 42.72 (88) | 57.28 (118) | <0.001 | 69.49 (82) | 30.51 (36) | NS |
| | Medium chance | 46.09 (112) | 53.91 (131) | | 65.65 (86) | 34.35 (45) | |
| | High chance | 22.44 (35) | 77.56 (121) | | 59.50 (72) | 40.50 (49) | |
| How severe do you think it would be if you get COVID-19? | No or low severe | 42.80 (104) | 57.20 (139) | <0.001 | 66.91 (93) | 33.09 (46) | NS |
| | medium severe | 45.91 (101) | 54.09 (119) | | 66.39 (79) | 33.61 (40) | |
| | Very severe | 21.13 (30) | 78.87 (112) | | 60.71 (68) | 39.29 (44) | |
| We need to prioritize going back to our normal routines (opening schools, colleges, offices) as soon as possible by maintaining safety protocols. | Agree | 42.79 (184) | 57.21 (246) | 0.01 | 64.63 (159) | 35.37 (87) | NS |
| | Not sure | 29.03 (27) | 70.97 (66) | | 65.15 (43) | 34.85 (23) | |
| | Disagree | 29.27 (24) | 70.73 (58) | | 65.52 (38) | 34.48 (20) | |
| Have you heard of any vaccine (s) that have been approved globally for mass inoculation? | No | 43.75 (189) | 56.25 (243) | <0.001 | 62.14 (151) | 37.86 (92) | NS |
| | Yes | 26.59 (46) | 73.41 (127) | | 70.08 (89) | 29.92 (38) | |
| The vaccines that are being inoculated worldwide are effective and safe | Agree | 27.07 (62) | 72.93 (167) | <0.001 | 47.90 (80) | 52.10 (87) | 0.003 |
| | Neutral | 33.47 (79) | 66.53 (157) | | 75.80 (119) | 24.20 (38) | |
| | Disagree | 67.14 (94) | 32.86 (46) | | 89.13 (41) | 10.87 (5) | |
| Vaccines should be marketed and distributed entirely by the government- what do you think? | Agree | 39.02 (199) | 60.98 (311) | NS | 63.34 (197) | 36.66 (114) | NS |
| | Neutral | 43.28 (29) | 56.72 (38) | | 78.95 (30) | 21.05 (8) | |
| | Disagree | 25.00 (7) | 75.00 (21) | | 61.90 (13) | 38.10 (8) | |

(*Continued*)

**Table 4.** (Continued)

| Variable | Levels | Willingness to be vaccinated | | | Willingness to immediate vaccination | | |
|---|---|---|---|---|---|---|---|
| | | No % (n) | Yes % (n) | P-value | delay % (n) | immediate % (n) | P-value |
| If I decided to get the COVID-19 vaccine, it would be hard to find a provider or clinic that could give me the vaccine. | Agree | 26.56 (68) | 73.44 (188) | <0.001 | 60.11 (113) | 39.89 (75) | 0.02 |
| | Not sure | 27.93 (50) | 72.07 (129) | | 74.42 (96) | 25.58 (33) | |
| | Disagree | 68.82 (117) | 31.18 (53) | | 58.49 (31) | 41.51 (22) | |
| The COVID-19 vaccine might have side effects. | Agree | 78.29 (101) | 21.71 (28) | <0.001 | 67.86 (19) | 32.14 (9) | NS |
| | Not sure | 32.33 (75) | 67.67 (157) | | 61.78 (97) | 38.22 (60) | |
| | Disagree | 24.18 (59) | 75.82 (185) | | 67.03 (124) | 32.97 (61) | |
| Considering the current scenarios, who do you think should receive the first shipment of the vaccine in Bangladesh? | Healthcare workers/ professionals | 42.42 (154) | 57.58 (209) | NS | 60.77 (127) | 39.23 (82) | NS |
| | Elderly people (60+ years) | 33.33 (23) | 66.67 (46) | | 65.22 (30) | 34.78 (30) | |
| | people who have underlying diseases | 36.21 (21) | 63.79 (37) | | 64.86 (24) | 23.88 (16) | |
| | Politicians | 33.00 (33) | 67.00 (67) | | 76.12 (51) | 16.42 (11) | |
| | Other (specify) | 26.67 (4) | 73.33 (11) | | 72.73 (8) | 27.27 (3) | |
| Knowledge about COVID-19 and COVID-19 vaccination | Limited knowledge | 89.90 (89) | 10.10 (10) | <0.001 | 60.00 (6) | 40.00 (4) | NS |
| | Medium knowledge | 38.72 (103) | 61.28 (163) | | 63.80 (104) | 36.20 (59) | |
| | Good knowledge | 17.92 (43) | 82.08 (197) | | 65.99 (130) | 34.01 (67) | |

COVID-19 = coronavirus disease 2019; NS indicates a non significant result at 10% level.

being in the vaccine accept group compared to the respondents who believe if they get infected with COVID-19 it would be mild. The people with good knowledge about COVID-19 and COVID-19 vaccines exhibited 22.23 times higher odds of accepting the COVID-19 vaccine compare to people with lower knowledge about COVID-19 and COVID-19 vaccine.

The odds of accepting vaccines immediately was 47% lower among female respondents than male. The odds of accepting the COVID-19 vaccine immediately was 2.03 times higher among rural respondents compared to urban respondents. Hindu participants displayed 3.79 times higher odds of taking the COVID-19 vaccine immediately compared to Muslim participants. The odds of taking the COVID-19 vaccine immediately was 84% lower for the respondents who heard about any vaccine (s) that have been approved globally for mass inoculation compare to respondents who did not hear about any vaccine (s) that have been approved globally for mass inoculation. Married people had almost 2 times higher odds of being vaccinated immediately compared to unmarried people. Participants who disagreed with the statement *"the vaccines that are being inoculated worldwide are effective and safe"* had a 68% lower likelihood to accept the COVID-19 vaccine immediately compared to those who agreed with it.

## Discussion

The vaccine confidence in the public would be lower because of the uncertainties of new vaccines and new infectious diseases [26]. Although estimates of herd immunity and vaccination are changing rapidly, some of the estimates indicated at least 60% of a population needs to be vaccinated to achieve herd immunity. To ensure equitable distribution of the COVID-19 vaccines, it is crucial to make a projection of the acceptance in public and identify the predictors associated with vaccine acceptance [21, 27]. However, without any projection, in Bangladesh, a nationwide COVID-19 vaccination campaign started on February 7, 2021, and 0.26% of people were registered to receive the COVID-19 vaccine and 0.1% were vaccinated till February 27, 2021 [28]. On March 24 the dose administration rate in Bangladesh was 3%. Although this is a good achievement for Bangladesh, this vaccination rate is still lower than many countries such as UK (46%), United States (38%), Maldives (41%), India (3.6%) [29].

The findings of this study showed that a large proportion of respondents wanted to get vaccinated and the acceptance rate nearly touched the threshold to achieve herd immunity (60%). However, the majority of them would delay their vaccination due to uncertainty of the safety and efficacy of the newly developed vaccine. We also found COVID-19 vaccine acceptability has a statistically significant correlation with socio-demographic characteristics such as age, gender, higher educational qualification, employment status. These findings are consistent with other researches conducted in the recent times in different countries. Determining the factors of acceptability of vaccine or immediate vaccination are complex and context-specific and the factors vary with time, place, and type of vaccines [30, 31]. In this study, the vaccine acceptability was higher among males, older, and highly educated people. Higher acceptability was also found among people who live in urban areas and have higher incomes. The socio-demographic factors were also found as significant factors for pandemic vaccine acceptability in the UK, France, Australia, the US, and Japan [21, 22, 32–34]. In Saudi Arabia, only age and marital status were found as significant factors in determining the willingness of accepting the COVID-19 vaccines [31]. We found that one of the strong correlates of vaccine acceptability was previous vaccination experience in adulthood. People who think they are at a higher risk of being infected with COVID -19, who believe that COVID-19 might be highly severe for them, and good knowledge about COVID-19 and COVID-19 vaccination were also found to be significantly linked with vaccine acceptability. The previous vaccination, vaccination beliefs, and attitudes about COVID-19 vaccination were also found as significant determinants in Italy, UK, China [21, 23, 35, 36]. Higher knowledge about COVID-19 symptoms, transmission routes, and prevention and control measures against COVID-19 were found associated with the willingness of accepting COVID-19 vaccine among the general population in Greece [37].

Vaccine safety has been reported as the main barrier for deciding for immediate vaccination among people especially for vaccines that are newly developed [32, 38, 39]. For instance, in a large vaccine accepted group (67%) in Australia, 13% of them wanted to delay in vaccination to see the efficacy and safety [7]. Several factors were identified in our study that are influencing the immediate vaccination intention among the public in Bangladesh. Gender, religion, region, and marital status were found to be associated with immediate vaccination. People who live in rural areas and males were more likely to be vaccinated immediately. The previous vaccination experience after the age of 18 was significantly associated with vaccine acceptance, though, it was not a significant factor for immediate vaccination which conflicts with other studies in different countries [7, 32, 33]. Employment status and knowledge about COVID-19 and COVID-19 vaccination were not significant in immediate vaccination decision-making among the Bangladeshi people.

Our results may be used to develop successful vaccination plans and immunization services for people who are afraid and/or hesitant of taking COVID-19 vaccines. To increase vaccine acceptability among rural people, baseless rumors and myths (especially on social media) against the COVID-19 vaccines must be checked and they should be reached out with scientific facts describing the safety and efficacy of the vaccines. To inspire females to get vaccinated, specialist doctors' opinions can be spread through social media, television, radio, and print media. General people and celebrities who have already taken the vaccine should share their experience (of no, mild, or severe side-effects) on social media and other mass media. Topics on infectious diseases, their preventive measures, and vaccination should be included in the textbooks to better prepare for future pandemics.

To the best of our knowledge, this is the first study to inquiry about the acceptability of the newly developed COVID-19 vaccine among Bangladeshi people. Our study divided participants based on the acceptance levels (immediate or delayed acceptance or refusal to accept) and provides associated factors that influence the vaccine acceptability and immediate vaccination. This study sheds light on the current scenarios of public attitudes and willingness regarding COVID-19 vaccination. Based on the findings, policymakers can identify the most priority groups (older than 70, rural, and women) or communities that need special attention in COVID-19 vaccination campaigns. Since a higher vaccination rate is the key factor to achieve herd immunity, mass people must be inspired to get vaccinated [40]. This research will help the policymakers make an effective vaccination strategy for a greater uptake rate of vaccines in a bid to control the COVID-19 pandemic.

However, this study has some limitations. Since the offline face-to-face survey is not possible during the COVID-19 pandemic, we have used the online platform to collect information that may limit the representativeness of the sample. We only reached out to those who had access to the internet and smart devices. A similar study was necessary before developing the COVID-19 vaccine to understand the changes in vaccination intention. Since self-reported information may lead to information bias, the findings of this study may differ from the real scenario. Further study is needed to inspect the changes in vaccination intention and its' determinants during the pandemic.

## Conclusion

A high prevalence of refusal and hesitancy about COVID-19 vaccination in Bangladesh was observed in the study. The safety concern seemed to be the main reason for the unwillingness to accept vaccines. To increase the immediate vaccination and vaccine acceptance rate among the public which is touted to be the best way to get rid of the devastating COVID-19 pandemic, vaccination campaigns need to be designed. Special emphasis should be given to inspire rural people, females, and senior citizens (70+). Besides, evidence-based communications and health education can reduce public vaccine hesitancy and concern about vaccine safety.

## Supporting information

**S1 Questionnaire. COVID-19 vaccine acceptability survey.**
(DOCX)

## Acknowledgments

We would like to express our deep and sincere gratitude to the respondents who took part in the study voluntarily and shared the link with others. We would also like to take this

opportunity to thank those who, despite not being eligible, shared the link and inspired others to participate.

## Author Contributions

**Conceptualization:** Sultan Mahmud, Md. Mohsin.

**Data curation:** Sultan Mahmud, Md. Mohsin, Ijaz Ahmed Khan, Ashraf Uddin Mian, Miah Akib Zaman.

**Formal analysis:** Sultan Mahmud, Md. Mohsin, Ijaz Ahmed Khan, Ashraf Uddin Mian.

**Investigation:** Sultan Mahmud, Md. Mohsin.

**Methodology:** Sultan Mahmud, Md. Mohsin.

**Project administration:** Sultan Mahmud.

**Software:** Sultan Mahmud.

**Supervision:** Sultan Mahmud.

**Validation:** Sultan Mahmud, Md. Mohsin.

**Visualization:** Sultan Mahmud, Md. Mohsin, Ijaz Ahmed Khan, Ashraf Uddin Mian.

**Writing – original draft:** Sultan Mahmud, Md. Mohsin, Ijaz Ahmed Khan, Ashraf Uddin Mian, Miah Akib Zaman.

**Writing – review & editing:** Sultan Mahmud, Md. Mohsin, Ijaz Ahmed Khan, Ashraf Uddin Mian, Miah Akib Zaman.

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
