## [Decision Letter · Decision Letter 0]

9 Aug 2021

PONE-D-21-11048

Acceptance of COVID-19 Vaccine and Its Determinants in Bangladesh

PLOS ONE

Dear Dr. Mahmud,

Thank you for submitting your manuscript to PLOS ONE. After careful consideration, we feel that it has merit but does not fully meet PLOS ONE’s publication criteria as it currently stands. Therefore, we invite you to submit a revised version of the manuscript that addresses the points raised during the review process.

Your manuscript has undergone the peer-review process and the reviewers have provided their comments/suggestions. Kindly address these points/concerns before we make a decision.

We look forward to receiving your revised manuscript.

Kind regards,

Kingston Rajiah

Academic Editor

PLOS ONE

Journal Requirements:

3. Please revise your tables to replace p-values of "0.000" to "<0.001.

Reviewers' comments:

Reviewer's Responses to Questions

**Comments to the Author**

1. Is the manuscript technically sound, and do the data support the conclusions?

Reviewer #1: Yes

Reviewer #2: Yes

2. Has the statistical analysis been performed appropriately and rigorously? 

Reviewer #1: Yes

Reviewer #2: Yes

3. Have the authors made all data underlying the findings in their manuscript fully available?

Reviewer #1: Yes

Reviewer #2: Yes

4. Is the manuscript presented in an intelligible fashion and written in standard English?

Reviewer #1: Yes

Reviewer #2: Yes

5. Review Comments to the Author

Reviewer #1: Dear author,

Please add population size, age range & ethical approval number from organization in the abstract

Concise the introduction its too lengthy

There is no need to mention results again in discussion only discuss your results with previous studies

Make the article more concise and compact

Reviewer #2: The study performed is current with sound idea and findings. The methodology and results are well presented in manuscript. The work is well appreciated for collection and presentation of data. The title is very simple, its must be elaborated.

6. PLOS authors have the option to publish the peer review history of their article (what does this mean?). If published, this will include your full peer review and any attached files.

Reviewer #1: No

Reviewer #2: No

---

## [Author Response · Author response to Decision Letter 0]

21 Aug 2021

August 21, 2021

Editor in Chief

Emily Chenette

PLOS ONE

Subject: Submitting Revised Manuscript [PONE-D-21-11048] for the journal “PLOS ONE”

Title: Knowledge, beliefs, attitudes and perceived risk about COVID-19 vaccine and determinants of COVID-19 vaccine acceptance in Bangladesh

Dear Editor,

Thank you for allowing me to resubmit the revised manuscript. We appreciate the time and effort that you and the reviewers have dedicated to providing us with your valuable feedback on our manuscript. We are truly grateful to the reviewers for their insightful comments on our paper. We believe these suggestions will help improve the quality of the manuscript. We have been able to incorporate changes to reflect all the suggestions provided by the reviewers. We have highlighted the changes within the manuscript by using Track Changes. We have also provided an unmarked version of our revised paper without tracked changes.

Here is the point by point responses to the reviewers’ comments and concerns: 

Response to Editor: 

Editor point #1: “Please ensure that your manuscript meets PLOS ONE's style requirements, including those for file naming. The PLOS ONE style templates can be found at 

https://journals.plos.org/plosone/s/file?id=ba62/PLOSOne_formatting_sample_title_authors_affiliations.pd”

Author response #1: We would like to thank you for your suggestion. We have separated the main body and the title pages of the manuscript and these two files have been organized based on the instructions. 

Editor point #2: “Please review your reference list to ensure that it is complete and correct. If you have cited papers that have been retracted, please include the rationale for doing so in the manuscript text, or remove these references and replace them with relevant current references. Any changes to the reference list should be mentioned in the rebuttal letter that accompanies your revised manuscript. If you need to cite a retracted article, indicate the article’s retracted status in the References list and also include a citation and full reference for the retraction notice.”

Author response #2: Thank you for the legit suggestion. We have rechecked the reference list it is complete and correct. 

Editor point #3: “Please revise your tables to replace p-values of "0.000" to "<0.001”

Author response #3: Thank you for the suggestion. It has been done.

Editor point #4: “Thank you for stating the following financial disclosure: 

Please include your amended statements within your cover letter; we will change the online submission form on your behalf.”

Author response #4: Since we did not receive any funding for this study, we have stated “The authors received no specific funding for this work.”

Response to Reviewer #1: 

Reviewer point #1: “Dear author,

Please add population size, age range & ethical approval number from organization in the abstract 

Concise the introduction its too lengthy 

There is no need to mention results again in discussion only discuss your results with previous studies 

Make the article more concise and compact”

Author response #1: We would like to thank you for these detailed comments. We have added population size, age range, and details regarding ethical approval in the abstract. We have refurbished the introduction and discussion to make them more succinct. We have removed the redundant parts from the discussion section. 

Response to Reviewer #2: 

Reviewer point #1: “The study performed is current with sound idea and findings. The methodology and results are well presented in manuscript. The work is well appreciated for collection and presentation of data. The title is very simple, its must be elaborated.”

Author response #1: Thank you for your appreciation and encouraging words. As you suggested we have elaborated the title of this manuscript. 

In addition, we went through the whole article for a similar types of typos and editing mistakes. We believe that the changes have improved the revised manuscript, which you will find updated.

---

## [Editor Report · Decision Letter 1]

24 Aug 2021

Knowledge, beliefs, attitudes and perceived risk about COVID-19 vaccine and determinants of COVID-19 vaccine acceptance in Bangladesh

PONE-D-21-11048R1

Dear Dr. Mahmud,

We’re pleased to inform you that your manuscript has been judged scientifically suitable for publication and will be formally accepted for publication once it meets all outstanding technical requirements.

Kind regards,

Kingston Rajiah

Academic Editor

PLOS ONE
---

## [Editor Report · Acceptance letter]

31 Aug 2021

PONE-D-21-11048R1 

Knowledge, beliefs, attitudes and perceived risk about COVID-19 vaccine and determinants of COVID-19 vaccine acceptance in Bangladesh 

Dear Dr. Mahmud:

I'm pleased to inform you that your manuscript has been deemed suitable for publication in PLOS ONE. Congratulations! Your manuscript is now with our production department. 

Kind regards, 

on behalf of

Dr. Kingston Rajiah 

Academic Editor

PLOS ONE